# The Near-Infrared Spectroscopy of Ethanol-Fixed Tissues to Detect Illicit Treatments with Glucocorticoids in Bulls

**DOI:** 10.3390/foods11193001

**Published:** 2022-09-27

**Authors:** Salvatore Barbera, Giorgio Masoero, Carlo Nebbia

**Affiliations:** 1Dipartimento di Scienze Agrarie, Forestali e Alimentari, Università di Torino, Largo Paolo Braccini, 2-10095 Grugliasco, Italy; 2Accademia di Agricoltura di Torino, Via A. Doria, 10-10132 Torino, Italy; 3Dipartimento di Scienze Veterinarie, Università di Torino, Largo Paolo Braccini, 2-10095 Grugliasco, Italy

**Keywords:** UV-Vis-NIR spectroscopy, beef, illicit growth promoters, dexamethasone, ethanol fixing, *thymus*

## Abstract

This study aimed to set up indirect, rapid methods involving near infrared (NIR) spectroscopy analysis, to detect illicit treatments with glucocorticoids in bull. The ethanol fixation method (EtOH) was applied to 7 different tissues obtained from 20 Friesian bulls, 12 of which were experimentally administered with dexamethasone as part of a growth-promoting protocol for 60 days and slaughtered 26 days after the end of the treatment. A perfect discrimination was obtained for the 7 sampled tissues, considering a full UV-Vis-NIR range (350 ÷ 2500 nm), for both false positive and negative animals. The validated true positive and negative errors were zero for the *longissimus thoracis* muscle, 10% for the skin-*dermis*, 15% for the fat, 25% for the *thymus* gland and the *semitendinosus* muscle, 30% for the *sternomandibularis* muscle and 35% for the skin-hair. A multiple test on the most accessible tissues, that is, the *thymus* gland, the *sternomandibularis* muscle and fat, can be used as an alternative to provide indications about animals that have been subjected to illicit treatments. In the short space of three days from the slaughter, NIR spectroscopy of ETOH fixed tissues, would allow at least cost the detection of a probable illicit which could eventually be reported to health authorities for specific investigation in the frame of official controls.

## 1. Introduction

Near-infrared spectroscopy (NIRS) provides complex chemical and physical information linked to the vibration behavior of molecular bonds such as C-H, O-H, and N-H [1]. Since fiber optics instruments have reached an efficient degree of portability, several researchers have applied this technique to either raw or transformed meat products, at-line or on-line [2,3,4,5] for both food safety evaluation and control purposes [1,6,7].

NIRS has been used extensively as a rapid, cost-effective, and eco-friendly method. However, the abundance of water in lean meat species and the low level of energy applied in the measuring process are well-known limitations of the application of NIRS to meat samples. To circumvent these problems, samples are treated and fixed with ethanol (EtOH) solutions, a process that induces partial dehydration and protein coagulation. This kind of sample treatment has been applied successfully to investigate, among others, ontogenic, genetic and/or nutritional effects in cattle [8], buffaloes [9], pigs [10], rabbits [11] and salmon [12].

The illegal use of synthetic glucocorticoids (GC) for growth-promoting purposes is a relatively common practice in meat cattle breeding. Such drugs, administered alone or in combination with other substances (i.e., steroid hormones and β-agonists), can improve the productive performance of animals, with economic benefits for the farmers. However, since GC residues in animal products are potentially dangerous for the consumer, very low maximum residue limits have been set for GC in bovine edible tissues in Europe [13]. The official control for GC abuse is based on the chemical analysis of the drug residues in urines or tissues; however, GC elimination in cattle is very rapid, and complete depletion already occurs few days after treatment cessation.

Only limited applications of NIRS in the field of food safety have been reported [1], including the analysis of meat samples from bovines administered with illegal growth promoters [14]. The present work was aimed at exploring the ability of vibrational spectroscopy to detect permanent changes in muscles and in other EtOH-fixed tissues from cattle treated with dexamethasone, a synthetic GC that is widely used as an illicit growth promoter in beef cattle. Unlike a similar study [14], tissues were sampled well after treatment withdrawal (26 days). Interestingly, changes in meat composition and meat characteristics were reported in a companion study conducted on the same animals from this trial [15].

Our study was thus performed to aid both food safety and food traceability. In view of a possible standardization of the technique, sampling was performed on tissues readily available at slaughtering (referred to as “easy”). For the sake of simplification and rapidity of operation, the *sternomandibularis* muscle was sampled. Measurements were also conducted on the *thymus* gland, which is the target organ for the histological screening of GC abuse in bovine species [16], as well as on the external and the internal sides of skin sampled from the head, and on the abdominal fat.

A wide range spectrum UV-Vis-NIR instrument and three radiation intervals were evaluated in this study to define the possibility of using different bench and portable instruments.

## 2. Materials and Methods

### 2.1. Animal Treatment, Sample Collection and Preparation

The study was carried out on tissue samples obtained in a previous experiment, as described in detail by Barbera et al. [15]. Briefly, 20 Italian Friesian finishing bulls (average initial weight 440 ± 23 kg) were allotted to 3 groups and were daily administered 0 (group N, n = 8), 0.7 (n = 6), or 1.4 (n = 6) mg of dexamethasone sodium phosphate/head/day for 60 days. Because of the small number of experimental subjects, the animals from the two dexamethasone-treated groups were mixed to form a single group (T) for the spectral elaborations. The used schedule was intended to mimic a growth-promoting protocol [17]. The animals were slaughtered 26 days after the end of the treatment, and carcasses were kept at 0–4 °C for 24 h. Small specimens of five easily accessible tissues were then collected from the *thymus* gland, *sternomandibularis* muscle (MSM), and from the fat; skin (hair and *dermis*) was sampled immediately after slaughtering. Similar specimens were also collected from more representative, but difficult to sample, “noble” muscles, i.e., *longissimus thoracis* (MLT) and *semitendinosus* (MST), hereafter referred to as “less accessible”. The muscle, *thymus* gland and fat specimens were excised using a 1-inch Ø die cutter, for lengths of about 5 cm, and introduced into a 30 mm Ø * 115 mm, 50 mL Falcon tube, which was then filled with 95% ethanol (EtOH) and kept at +4 °C, for 24 h.

### 2.2. Data Acquisition

The pre-scanning processing consisted of a static aeration of the specimens over filter paper for 2 h. A total of seven different specimens were prepared: three different muscles (MSM, MLT, MST), the fat, the *thymus* gland, the hair (external part of the skin) and the *dermis* (internal part of the skin). A portable UV-Vis-NIR spectrometer (Model LSP 350-2500P, LabSpec Pro ASD, Analytical Spectral Devices Inc., Boulder, CO, USA), equipped with fiber optics to collect spectra from 350 to 2500 nm, was used to scan the samples.

The whole procedure (from slaughtering to the acquisition and evaluation of NIRS data) required about three days.

### 2.3. Statistical Analysis

Each spectrum was taken as the average of 25 scans. The 2151 reflectance points (R), transformed as Log (1/R), were analyzed in principal component analysis (PCA) to shape the objects in a bi-plan, using XLStat 2019.4.1 (Addinsoft). PCA is a means to reduce the high number of vibrational variables to a smaller number of latent variables, thereby limiting the loss of information as much as possible. For the classification purposes, the spectra were then imported into SAS V.9 software to conduct a discriminant analysis, with ONE (leave-one-out) cross-validation.

The whole spectral set of wide UV-Vis-NIR radiation was then divided into three subranges, corresponding to minor classes of vibrational instruments, namely: “cut-off a” (UV-Vis-NIR range, 350–1025 nm), corresponding to a simple Vis-NIR instrument; “cut-off b” (NIR range, 740–1070 nm), corresponding to a low-cost spectrometry system for end-user food analysis NIR-SCiO^TM^ [18]; “cut-off c” (UV-Vis range, 350–681 nm), corresponding to image analyzers.

To show the action of the illicit treatment, the difference between treated and control, expressed as percent of control [(T − N)/N × 100], was calculated to show any treatment-related change in absorbance.

The error rate statistics of the treated (T, false−) and control (N, false+) animals were assessed by means of Medcalc software [19] for testing proportions. The error rates in the calibration and cross-validation modes were considered independently for the seven tissues. However, a chaining of the test, repeated for each animal, can also be applied. Thus, progressive compound probabilities (CP) were considered by joining together the results of the test errors (P) from the classifications of the different tissues of each animal [20]. For example, for a T animal, for which a positive result had been obtained from two tests with *p* = 0.4 errors, the CP of the chained test was 0.4 × 0.4 = 0.16, which accounted for the probability of that T animal obtaining the same error score in the two tests; a third positive test with *p* = 0.4 also gave an added CP = 0.16 × 0.4 = 0.064. In general, only one out of 16 T animals (16 = 1/CP) received three false negative scores after three tests with probability 0.4 each.

## 3. Results

### 3.1. NIR Spectra

The average spectra of the tissues (T and N together) appeared to be different (Figure 1a). Only samples from the three muscles (MSM, MTL, MST) showed similar shapes over the entire wavelength range, while those from other tissues, and the hair in particular, looked different.

In general, the treatment induced a higher reflectance in the tissues. Consequently, difference between treated and control [(T − N)/N × 100] were on average below 8.2%, with an average and constant gap of −23% for the *thymus* gland, as shown in Figure 1b.

### 3.2. Principal Component Projections

The principal component projection on the first two axes on average featured a variability of 91.7% (Figure 2).

As far as the MLT, the *thymus* gland, fat and hair were concerned, the T and N objects showed distinct patterns with similar shapes for MST and hair. More uniform distributions were instead observed for the MSM and *dermis* samples

### 3.3. Discriminant Analysis

Linear discriminant analyses, a statistical function that is useful for determining whether a set of continuous variables is effective in predicting category membership, was applied to all acquired spectra. In calibration mode, the results achieved total discrimination of all the considered tissues over all the spectral ranges.

When classifying a “new” animal (Table 1) as treated (T) or untreated (N), an average misclassification rate of 0.20 (total error) over the entire spectral range was seen, although tissue-related different error rates were observed. While no total error was observed for MLT (*p* < 0.0001), the error rate was 0.10 for the *dermis* (*p* = 0.0003), 0.15 for the fat (*p* = 0.0017), and 0.25 for the *thymus* gland and MST (*p* = 0.0253), respectively. For MSM was detected a 0.30 error rate, while 0.35 was observed for the hair (not significant). Converting the error rate into specificity (ability to correctly identify the untreated samples) and sensitivity (ability to correctly identify the treated samples) percentages, the sum of the seven scores averaged 0.75 for the specificity and 0.83 for the sensitivity, two highly significant, but not statistically different values (*p* = 0.2498).

To establish the discriminatory ability of the method, the vibrational range was divided into three cut-off segments (Table 2), each corresponding to a different class of commercial instruments.

The seven different tissues and two different assessments (specificity f+, sensitivity f−) were considered.

As regards the overall false rate (negative for the treated group and positive for the control one), minimum errors were observed as mean rates for MST (0.05) and for the *thymus* gland (0.06), although statistical significance was reached for all the considered ranges. MLT ranked first for “cut-off a” but did not rank so well for “cut-off b” or “cut-off c”. The hair and the MSM excelled in the UV-Vis range, where, overall, the lowest average misclassification rates (0.25 and 0.14) were observed, with 8 significant rates (out of the fourteen considered ones). The least efficient range was that of the short NIR (740–1070 nm), with only 3 significant rates when applied to the 3 most easily accessible tissues (fat, MSM, *thymus*) (Table 3).

As far as the total errors (combined sum) are concerned, after combining the fat (f− = 0.42, Table 2 b. NIR), MSM (f− = 0.58) and *thymus* (f− = 0.08) values, for the 12 treated animals, error rates of 0.08, 0.25 and 0.33 were observed for the triple, double, and single f− test, respectively.

It should be noted that the observed 0.08 triple error rate f− was higher than the expected compound probability *p* = 0.019 derived from Π P (0.42 × 0.58 × 0.08). Conversely, the triple f+ error in 8 control animals was 0.00, a lower value than the expected 0.078.

## 4. Discussion

In our study, the application of NIRS to EtOH-fixed tissues from finishing bulls proved effective in detecting a treatment with dexamethasone that was used to mimic an illegal growth-promoting schedule that is widely applied under field conditions.

Interestingly, the detection capability of NIRS in discriminating treated from untreated tissues was also confirmed when using a different statistical approach (numeric regression, data not shown) from the discriminant analysis presented here [21]. NIRS techniques are used extensively, although mainly for meat quality control purposes [2,3], but very few applications have been reported so far for safety purposes involving animals exposed to illicit drug treatments or contaminants [1].

Berzaghi et al. [14] investigated the ability of spectroscopy to identify illegal treatments in beef cattle that had been administered dexamethasone following a similar protocol to that adopted in this study, but NIRS was applied to minced or sliced, non-EtOH-treated, *longissimus thoracis* samples. When the meat samples were subjected to discriminant analysis, followed by cross validation, about 63% of the controls and only 39% of the samples from the treated animals were correctly classified, that is, poorer results than the zero errors found for the same muscle in the present work. In addition, good discrimination ability was found in our study for a much longer time after treatment withdrawal and also for non-muscle samples.

In our opinion, sample preparation with EtOH could play a significant role in the successful discrimination between treated and untreated animals when using NIRS techniques. As a general consideration, it is possible to state that this procedure can be considered the basic process to obtain partially dehydrated specimens.

Vibrational spectroscopy can be used to appreciate several traits in EtOH preparations that do not appear in ordinary raw or minced tissues, mainly because of the strong absorption of radiation from H-OH but also because of some denaturation mechanisms in the tissue matrix. As can be seen in Figure 1b, the loss of absorption of the samples from the T animals is remarkable. This pattern points to a general paucity of substances that can capture radiation. However, it could also simply be the result of a relative abundance of water in the T animals. In the companion paper performed on the same animals examined in this study [15], no significant variations in the moisture content of the raw MLT were observed in treated bulls; on the other hand, the red index and chroma color parameters increased by 5%, and dramatic changes were observed for the rheological properties, with a 54% increase in cooking loss and 25% in meat tenderness, which would seem to indicate a different pattern of water distribution and mobility in the muscles [16,22]. With the exception of β-agonists, illegal growth-promoting treatments of meat-producing species are generally associated with: (i) an increase in lean meat; (ii) some retention of water in the carcass; (iii) an increase in meat tenderness [23]. The spectroscopy of EtOH-fixed tissues is depending on the latter two effects, possibly resulting in a treatment-related fingerprint of the muscles. However, in our study, the dexamethasone-related changes also involved the *thymus* gland and the skin and were apparent in the fat. In this respect, it should be noted that the prolonged administration of dexamethasone at growth-promoting dosages has been reported to cause *thymus* atrophy, due to a progressive replacement with adipose tissue [16], and an increase in the thickness of the external fat [24].

The color parameters of the EtOH-fixed tissues are thus concerned to a great extent, and a general pale shade distinguished the samples taken from the treated (“bloated”) animals from the control ones.

The administration of dexamethasone according to a growth-promoting schedule resulted in changes in both the transcriptomic [25] and proteomic [26] profiles of the muscle as well as in the meat quality traits [15,23,27].

Transcriptional biomarkers and immunohistochemistry have been found to be highly effective in the detection of illicit dexamethasone administration in veal calves (although not in bulls as in the present work) and to provide a cross validated error rate of about 12–14% [28].

As far as vibrational instruments are concerned, the main relationships are mostly concentrated in the UV-Vis region. In fact, over the seven examined tissues, the full 350–2500 nm range (Table 1) expressed three T false- values and one N false+ significant value; when NIR radiation was excluded, the significant results increased to five T false− and three N false+ values, respectively (Table 2).

In consideration of the above results, UV-Vis spectrometer instruments may be preferable to smart NIRS instruments (e.g., SCiO^TM^) for use in off-line tests on highly reputed EtOH-fixed noble meat samples, such as MST samples. However, based on the results shown in Table 1, it would be possible to “*a priori*” calculate what the compound probability of obtaining multiple responses would be, according to the used instrument. For instance, a T animal that was found positive with a “cut-off b” instrument for the *thymus* (*p* = 0.08, Table 3), the fat (*p* = 0.42) and the MSM (*p* = 0.58), would have a compound misclassification probability of 0.019 of being false, and, in fact, the observed error rate was 0.028 (Table 3).

The results of these studies also support the possible use of vibrational spectroscopy to disclose drug-induced changes in edible and non-edible tissues from illegally treated finishing bulls, thereby acting as a useful complement to chemical analysis. In this re-spect, “omics” techniques have been proposed [25,26,28,29] and multiresidue analytical methods are currently being implemented to detect anabolic substances, not only in tissues but also in biological fluids (e.g., bovine bile and urine) [30]. However, such methods are time consuming and expensive and are unlikely to be used as large-scale screening methods. In addition, all GCs are characterized by a rapid urinary elimination making their chemical detection unfeasible as early as few days after slaughtering [13].

## 5. Conclusions

As reported by Kademi, Ulusoy, and Hecer [31], spectroscopic devices may be used as a promising alternative to large, expensive, and complex types of devices for food analysis, inspection, and control purposes.

In the present work, we have used an indirect method (UV-Vis-NIR) to identify dexamethasone-induced changes in muscles and other organs in an authenticity test of a “legal-standard” product. The detection could be eventually reported to health authorities for specific investigations in the frame of official controls [27].

Based on our results, the spectroscopic examination of the ethanolic preparations of muscles taken from the first retail cuts, which are particularly difficult to collect, but also from secondary retail cuts or other organs and tissues that are instead easily available at slaughtering can be considered as an effective and efficient methodology. Even after a considerable time from treatment withdrawal, this method was able to identify animals illicitly treated with dexamethasone at growth-promoting dosages. To extend the method applicability, it would be necessary to establish a database that could be managed by a company’s self-control systems oriented to both food safety and traceability.

## Figures and Tables

**Figure 1 foods-11-03001-f001:**
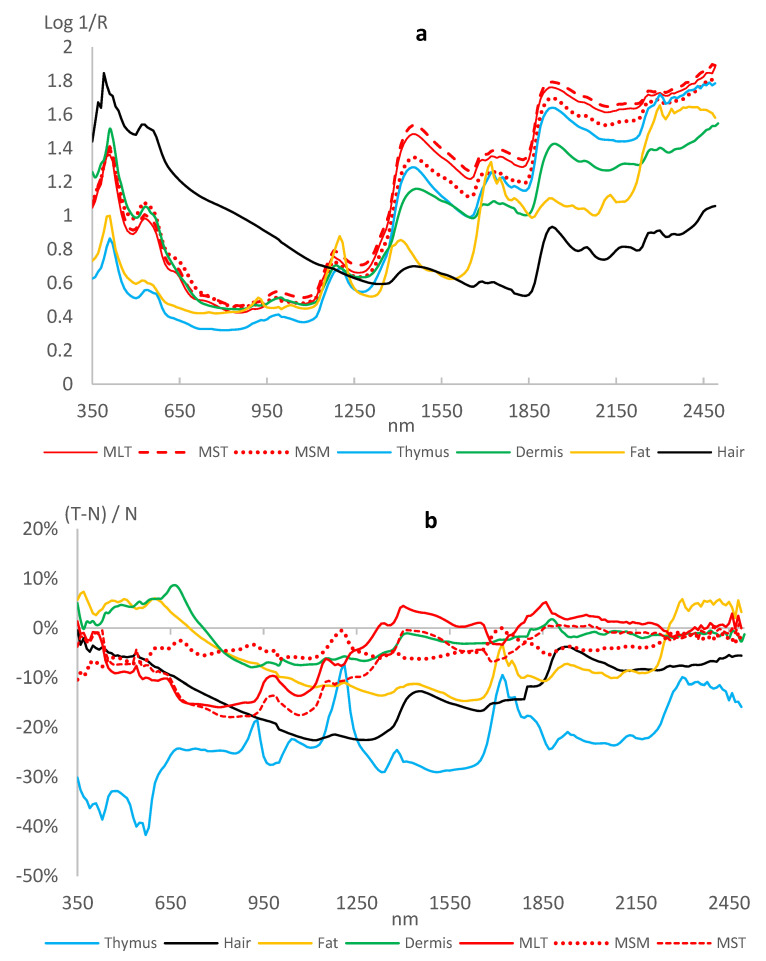
(**a**) Average of the full UV-Vis-NIR spectra of the seven tissues; (**b**) % of relative deviation in log(1/R) of the treated samples (T) with respect to the control ones (0% line) for the UV-Vis-NIR spectra of the seven tissues; MLT = *longissimus thoracis* muscle; MST = *semitendinosus* muscle; MSM = *sternomandibularis* muscle; Fat; *Thymus* gland; Hair; *Dermis*.

**Figure 2 foods-11-03001-f002:**
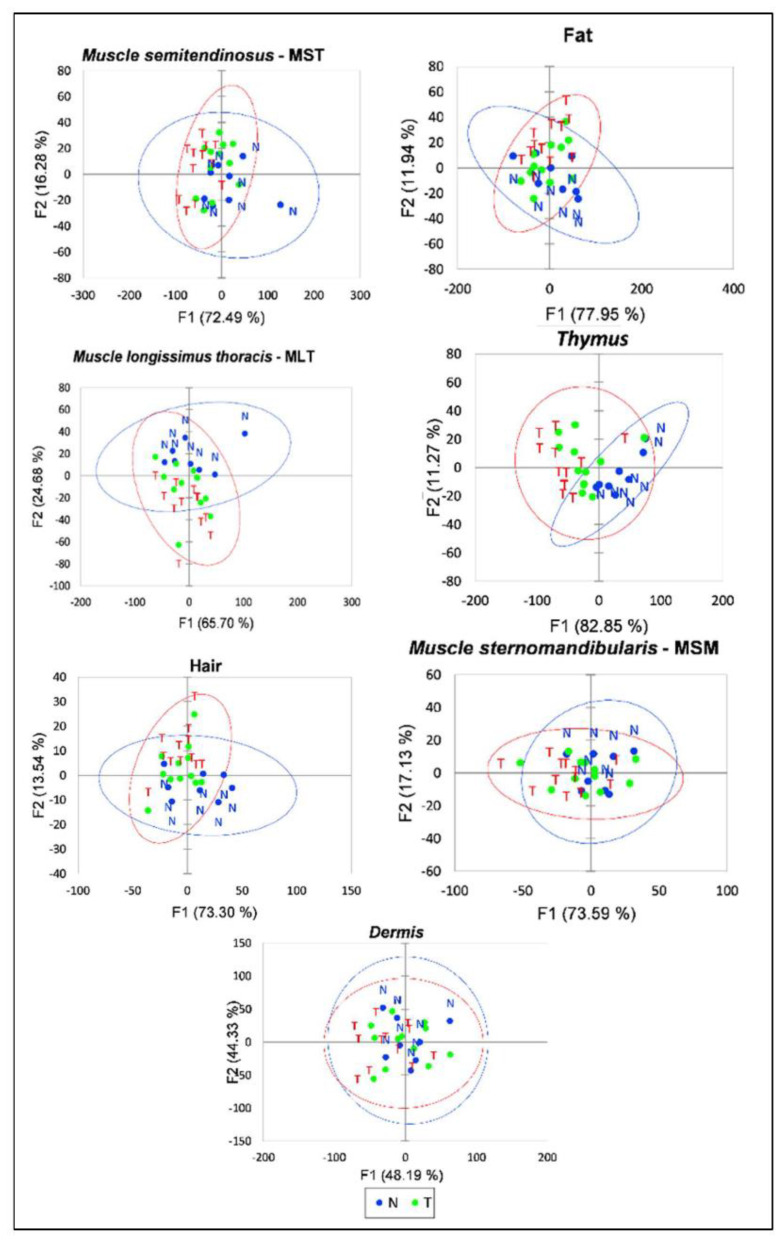
Principal component analyses of the full spectra for the seven tissues (N = Control; T = Treated).

**Table 1 foods-11-03001-t001:** Validated misclassification (as rate) in the control (N, false+, f+) and treated (T, false−, f−) animals for the seven tissues over the full spectral range (350 ÷ 2500 nm).

EtOH Tissue	N f+	T f−	Error Rate	Prob. Total
*Longissimus thoracis* muscle-MLT	0.00 **	0.00 **	0.00	0.0001
Skin-*Dermis*	0.25	0.00 **	0.10	0.0003
Fat	0.25	0.08 **	0.15	0.0017
*Thymus*	0.25	0.25+	0.25	0.0253
*Semitendinosus* muscle-MST	0.25	0.25+	0.25	0.0253
*Sternomandibularis* muscle-MSM	0.38	0.25+	0.30	0.0736
Skin-hair	0.38	0.33	0.35	0.1797
Total error			0.20	<0.0001
Specificity (1-false+)	0.75			0.0001
Sensitivity (1-false−)		0.83		<0.0001
Specificity vs. Sensitivity	0.75	0.83		0.2498

Significance vs. 50% threshold: + (*p* < 0.10); ** (*p* < 0.01).

**Table 2 foods-11-03001-t002:** Validated misclassification (as the rate) of the control (N, false+, f+) and treated (T, false−, f−) animals considering the seven tissues in three spectrum subranges of the “cut-off spectra”.

EtOH Tissue	Mean Rate	a. UV-Vis-NIR 350–1025 nm		b. NIR 740–1070 nm		c. UV-Vis 350–681 nm	
		N f+	T f−	N f+	T f−	N f+	T f−
*Semitendinosus* muscle-MST	0.05	0.25	0.00 **	0.00 **	0.00 **	0.13 *	0.00 **
*Thymus*	0.06	0.00 **	0.08 **	0.25	0.08 **	0.00 **	0.00 **
*Longissimus thoracis* muscle-MLT	0.24	0.00 **	0.00 **	0.50	0.25+	0.38	0.17 **
Skin–Hair	0.39	0.25	0.50	0.38	0.42	0.00 **	0.08 **
*Sternomandibularis* muscle-MSM	0.42	0.38	0.33	0.63	0.58	0.38	0.17 **
Skin-*Dermis*	0.46	0.50	0.50	0.25	0.25+	0.50	0.33
Fat	0.48	0.63	0.33	0.50	0.42	0.38	0.25+
Validated misclassification	Rate	0.29 **	0.25 **	0.36 *	0.29 **	0.25 **	0.14 **
False+ vs. False−	Prob.	0.6011		0.3863		0.1011	
Validated misclassification of the “cut-off spectra”		0.26 **		0.31 **		0.19 **	
Probability of contrasts	“Cut-off a”	1.000		0.3549		0.1615	
	“Cut-off b”			1.000		0.0206	

Significance vs. 50% threshold: + (*p* < 0.10); * (*p* < 0.05); ** (*p* < 0.01).

**Table 3 foods-11-03001-t003:** List of the validated classifications, with misclassifications indicated with *, of the twelve treated animals (T false−) and eight control animals (N false+) based on the three most accessible tissues in the “cut-off b” (740–1070 nm) spectrum subrange of the cut-off spectra, with the error rate of the combined tissues.

Animal	Treatment	Tissues			
		Fat (1)	MSM (2)	*Thymus* (3)	Combined Sum (1 & 2 & 3 *)
1	T	*	*	*	3
2	T				0
3	T	*	*		2
4	T				0
5	T				0
6	T	*	*		2
7	T		*		1
8	T				0
9	T	*	*		2
10	T		*		1
11	T	*			1
12	T		*		1
Error rate T False−		0.42	0.58	0.08	
* Misclassification in repeated tissues from 12 treated animals					
1 False−/12 = 0.33; 2 False−/12 = 0.25; 3 False−/12 = 0.08 0.11					
13	N	*		*	2
14	N	*			1
15	N			*	1
16	N		*		1
17	N		*		1
18	N	*	*		2
19	N		*		1
20	N	*	*		2
Error rate N False+		0.50	0.63	0.25	
* Misclassification in repeated tissues from 8 control animals					
1 False+/8 = 0.63; 2 False+/8 = 0.38; 3 False+/8 = 0.00					

***** Misclassified case; MSM muscle *Sternomandibularis*.

## Data Availability

Data is contained within the article.

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
