# Peer review of "The Near-Infrared Spectroscopy of Ethanol-Fixed Tissues to Detect Illicit Treatments with Glucocorticoids in Bulls"

_foods, 2022, doi:10.3390/foods11193001_

Round 1

Reviewer 1 Report

Dear Authors,

I find the topic interesting but please state the importance of the topic in the introduction section to show the relevance of the subject in terms of food science and technology. 

Indtroduction:

This is not a review article please remove Table 1

Please mention about the effects of treatments with dexamethasone, a glucocorticoid to animal tissues, and please explain t why there is a need for NIR methodologies to observe changes in tissues in terms of meat consumption (food safety? food traceability?)

Materails and Methods (presentation is poor)

Please divide this section into two subsections as 1. Materials 2. Methodology

Explain each method in detail with sub-titles. For example:

1. Data Acquisition

2. Statistical Analysis........

Results

Interpretation of the results is poor, please define the main outputs in detail and please make the font of graphs readible.

Discussion

Please discuss your study with other similar studies and please state the superiorites of your study when compared to previous ones.

Please consider a professional English correction

Author Response

Dear Reviewer,

thank you for your revision and we appreciate your suggestions and we hope to have answered to all your inquiries.

Response to Reviewer

In the revised version the introdiced changes are written in red and modified lines are indicated in the answer.

Point 1. This is not a review article please remove Table 1

R1. We agree, the Table 1 has been removed.

Point 2. Please mention about the effects of treatments with dexamethasone, a glucocorticoid to animal tissues, and please explain why there is a need for NIR methodologies to observe changes in tissues in terms of meat consumption (food safety? food traceability?).

R2. Lines 50-58, 68-74.

Both in the Intro and then in the Discussion sections, the effects of dexamethasone (growth-promoting schedule) on muscles and other tissues (e.g thymus) have been now more clearly described. Starting from the well-recognized ability of NIRS in detecting changes in tissue composition (e.g., water, fat content, etc), the use of NIRS is in our opinion recommended for food safety and food traceability purposes: according to our results, it would be able to discriminate between treated and untreated animals even long time (26 days) after treatment withdrawal. This would be a key feature particularly in the case of dexamethasone and other Glucocorticoids, which undergo a rapid elimination in the few days following treatment cessation. All these concepts have been introduced and/or highlighted in the Intro and in the Conclusion.

Point 3. Materials and Methods (presentation is poor)

Please divide this section into two subsections as 1. Materials 2. Methodology

Explain each method in detail with sub-titles. For example:

  1. Data Acquisition
  2. Statistical Analysis........

R3. The Material and Methods section has been modified to include the reviewer suggestions as follows:

  1. Animal treatment, sample collection and preparation
  2. Data acquisition
  3. Statistical Analysis

Moreover, we provided more details for PCA statistics.

Point 4. Results

Interpretation of the results is poor, please define the main outputs in detail and please make the font of graphs readible.

R4. Results have been reviewed.

To address the referee’s requests, results have described according to defined outputs (e.g., NIR spectra, Principal Component Descriptio, etc.). Text has been also changed to improve readability, and some non-explanatory graphs (Figure 1c and d) have been taken out. Graphs have been improved.

Point 5. Discussion

Please discuss your study with other similar studies and please state the superiorites of your study when compared to previous ones.

R5. Lines 237-243, 258-266 and other revisions along Discussion.

Despite the extensive application of NIRS to a variety of food preparations (including meat) only a few studies addressed the same issue of the present work. This was properly discussed in the Discussion section, when the superiorities of our study (discriminating ability also in non-muscle samples and after a much longer period after treatment withdrawal) were clearly stated.

Point 6. Please consider a professional English correction

R6. The text was revised by a native speaker with proficiency in Scientific English.

Reviewer 2 Report

In the manuscript foods-1878431 entitled “ NIR Spectroscopy of Ethanol-fixed Tissues to Detect Illicit Treatments with Glucocorticoids in Bulls”.  The study assesses rapid indirect methods to discover illicit treatments by Near Infrared Spectroscopy analysis. I have Major concerns in the attached PDF file; thus, I suggest the authors address those concerns in the revised version.

1- Please rewrite the title of the manuscript (Please, don’t begin with an abbreviation).

2- In the introduction, a lot of statements without the corresponding citation, please revise the citations of all statements.

3- Please describe all abbreviations in their first mention, it is preferable to add a list of abbreviations

4-  Please proofread the whole manuscript to avoid grammatical errors.

5- In the discussion section, please be more specific.

6- The conclusion section has to be shorter.

7- In the references section, please use journal style.  

Author Response

Dear Reviewer,

thank you for your revision and we appreciate your suggestions and we hope to have answered to all your inquiries.

Response to Reviewer

In the revised version the introdiced changes are written in red and modified lines are indicated in the answer.

Point 1. Please rewrite the title of the manuscript (Please, don’t begin with an abbreviation).

R1. The title has been rewritten.

Line 11 in the pdf – Abstract.

Abstract was revised as required by the note in the pdf.

Point 2. In the introduction, a lot of statements without the corresponding citation, please revise the citations of all statements.

R2. This has been done; Table 1 has been taken out and reviewed.

Point 3. Please describe all abbreviations in their first mention, it is preferable to add a list of abbreviations.

R3. All abbreviations are described at their first mention. A list of abbreviations has been added.

Point 4. Please proofread the whole manuscript to avoid grammatical errors.

R4. The whole manuscript has been proofread to avoid grammatical errors.

Lines 103-106 in the pdf - Statistics in Materials and Methods

Lines 109-124. More details have been added.

Line 155 in the pdf.

Line 168. Modified

Point 5. In the discussion section, please be more specific.

Line 176 in the pdf.

R5. The discussion was changed and the advantages of using NIRS over other indirect methods for the detection of illicit treatments have been duly emphasized.

Point 6. The conclusion section has to be shorter.

Line 241 in the pdf.

R6. The conclusion has been shortened as suggested.

Point 7. In the references section, please use journal style.

Line 287 in the pdf.

R7. The references were revised according to the journal style considering the available data. More recent references were added, and few old references were deleted.

Round 2

Reviewer 1 Report

The paper is acceptable with its revised form to be published in the  Journal of Food 

Reviewer 2 Report

All comments and suggestions done. Thanks